# How Much Information Does a Robot Need? Exploring the Benefits of Increased Sensory Range in a Simulated Crowd Navigation Task

**Marit Hagens and Serge Thill *** 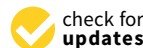

Donders Institute for Brain, Cognition, and Behaviour, Radboud University Nijmegen,
6525 XZ Nijmegen, The Netherlands; m.hagens@student.ru.nl

**\*** Correspondence: s.thill@donders.ru.nl

**Abstract:** Perfect information about an environment allows a robot to plan its actions optimally, but often requires significant investments into sensors and possibly infrastructure. In applications relevant to human–robot interaction, the environment is by definition dynamic and events close to the robot may be more relevant than distal ones. This suggests a non-trivial relationship between sensory sophistication on one hand, and task performance on the other. In this paper, we investigate this relationship in a simulated crowd navigation task. We use three different environments with unique characteristics that a crowd navigating robot might encounter and explore how the robot's sensor range correlates with performance in the navigation task. We find diminishing returns of increased range in our particular case, suggesting that task performance and sensory sophistication might follow non-trivial relationships and that increased sophistication on the sensor side does not necessarily equal a corresponding increase in performance. Although this result is a simple proof of concept, it illustrates the benefit of exploring the consequences of different hardware designs—rather than merely algorithmic choices—in simulation first. We also find surprisingly good performance in the navigation task, including a low number of collisions with simulated human agents, using a relatively simple A\*/NavMesh-based navigation strategy, which suggests that navigation strategies for robots in crowds need not always be sophisticated.

**Keywords:** crowd navigation; sensors; simulation

## 1. Introduction

Robots are increasingly playing societal roles, for example, they are already used in therapy for children with Autism Spectrum Disorder (ASD) [1–3], assistive robots in elderly (health)care [4,5] or guide robots at an airport [6]. All these robots require sensing abilities, whereas some also need the ability to navigate crowds of humans, such as the guide robot at an airport [6] or a social wheelchair robot for the elderly [5].

It is interesting to note, in this context, that in many robotics (and related) studies, similar problems are tackled with rather differing hardware set-ups. For crowd navigation, for example, even recent work in HRI uses a larger sensorised environment [7], but there are also examples of research using only on-board sensors (see, for example, [8]). This, in turn, raises the question of the added value of sensorising a certain environment compared to operating only based on local information.

In this paper, we are therefore interested in the trade-off between making increasingly global information (within the environment) available to the robot, and task performance. For convenience, we call the degree to which a robot can perceive global information of the environment "sensing sophistication", and in practical terms, we treat this simply as a distance from the location of the

robot. Low sensing sophistication thus corresponds to a robot that can only perceive its immediate surroundings, most likely using only on-board sensors (such as in [8]), whereas high sophistication would imply access to information that would often require additional sensors in the environment (such as in [7]). Note that the latter is not an infeasible proposition, even for large environments; in fact, much work in autonomous vehicles is concerned with solving navigation problems through cooperative systems that offer accurate information for large areas [9]. However, it is likely a more costly endeavour, both in the amount of sensors required, and the complexity of the algorithms required to process this information.

To the best of our knowledge, this question has received relatively little attention in the past. For some tasks, such as crowd navigation, it is possible to find—as mentioned above—studies that solve the solve the task either with local or global information. At a minimum, this hints at the fact that global information may not always be necessary for a successful implementation, but it is difficult to compare different studies in more detail since different set-ups, environments, specific tasks and so on render a comparison of performance meaningless. In many applications, it is also the case that current on-board sensors are often insufficient and external sensors therefore unavoidable. This is for example typically the case in robots used for ASD therapy [10]. Last, a comprehensive study of this trade-off would ideally require a range of different hardware set-ups available in the same task, which is often not feasible in practice given cost and resource constraints.

To address this last hurdle, we propose, in this paper, to use simulation environments to explore the trade-off between sensing sophistication and task performance. We choose to do so using a crowd navigation task, simply because previous research [7,8] suggests that it can be solved with both local and global information strategies. Our aim is a proof of concept; that is, we demonstrate the set-up for investigating this trade of in simulation environments but we are not aiming for a high-fidelity simulation, as a robot capable of navigating human crowds would, in practice, have to take into account human behaviour in its navigation strategy, which would require sophisticated models of such behaviour. For example, in addition to finding a path to a goal location, the robot would have to observe social conventions and respect safety concerns [11]. Previous research has found that a robot might end up fully immobile in complex crowd dynamics or environments, simply because it deems all possible actions unsafe [12]. A possible solution to this particular issue is to anticipate human cooperation [13], or to actively interact with humans [14]. Other challenges concern the optimality of the navigation, such as how to achieve collision-free navigation in a complex, and possibly unknown, environment with moving obstacles [15] or how to predict the likely path of a human being [16]. The models of human behaviour required to address such aspects in a satisfactory manner go beyond the scope of our present focus of evaluating task performance as a function of sensor sophistication.

The main question, therefore, is how a simulated robot's performance on a crowd navigation task scales with its sensor sophistication. We investigate this in simulations of different environments a crowd navigation robot might find itself in (an office corridor, an open space and an open space with static obstacles). Our main measure is the amount of time the robot requires to reach a goal n this environment while we also take into account the number of collisions with simulated humans it experiences. The main aim is to demonstrate that it is possible to cheaply run a number of such simulations to investigate a number of scenarios (environments and hardware configurations) at sufficient detail. Note that this remains a somewhat unconventional use of simulation; the far more typical approach being testing out different algorithms and strategies [11,17,18], but less so the exploration of different hardware strategies.

## 2. Methods

### 2.1. Simulation Environment

All simulations used in this paper are built in Unity (Unity Technologies, 2019), and simple GameObjects [19] are used to model humans and robots. Specifically, humans are modelled as objects

with height 2 and width 1 in world units, capable of moving at a speed of 3.5 world units/second and a maximal angular speed of 120 degrees per second. In all simulations, the robots are modelled with identical dimensions but have a reduced speed of 2.5 world units/second and maximal angular speed of 60 degrees/second.

## 2.2. Environments

We designed three different environments for this experiment. In each case, the distance between the robot and the goal is identical (although the actual path can vary in function of humans and obstacles). The starting point of the robot is fixed in each environment while human agents are placed randomly in each run. Human agents are programmed to navigate towards the starting point of the robot in the environment. As we populate the environment with a large number of humans, they will only start moving once the robot is within a distance of 20 world units.

These three environments are (Figure 1) as follows.

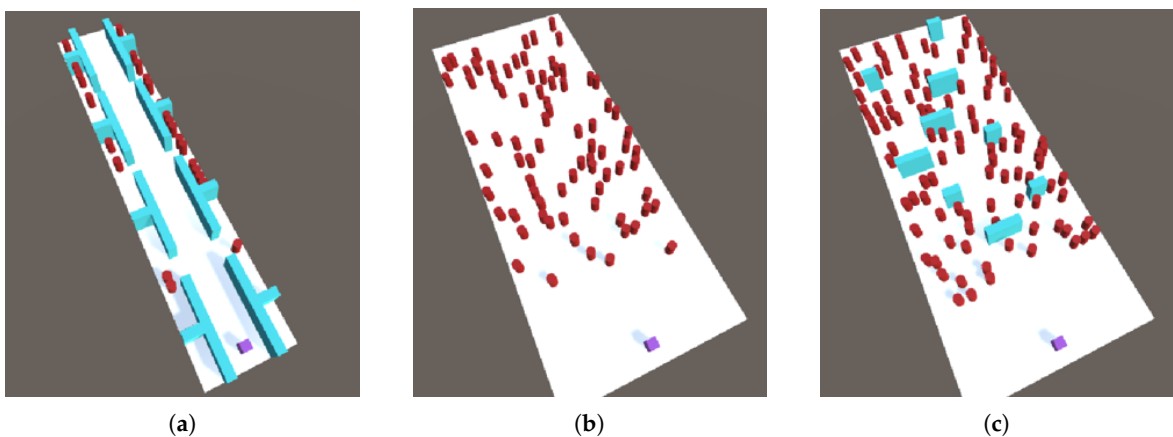

| (a) | (b) | (c) |

**Figure 1.** The three simulated environments used in this study: (**a**) office environment, (**b**) open street environment and (**c**) obstacles environment. Simulated human agents are shown as red cylinders, placed randomly at the start of a simulation.

- **Office environment**: Simulating a narrow office corridor that can accommodate at most three agents side-by-side. Offices are rooms accessible through doorways on either side of the corridor, and all human agents spawn inside these offices, preventing the robot from perceiving them until they enter the corridor.
- **Open street environment**: Simulating a large, open public square. Here, the environment offers no static obstacles that restrict the movement of the agents.
- **Obstacles environment**: Simulating an open environment that contains a number of obstacles in the form of static objects.

## 2.3. Pathfinding Algorithms

All agents (humans and the robot) navigate using A* search on a NavMesh. Humans may additionally move sideways with a small probability of 0.1, rendering their movements less predictable. A* search is widely used to find the shortest path between two points, combining uniform-cost search and heuristic search to repeatedly explore the most rewarding unexplored locations [20]. To create the required locations, we use Navigation Mesh (NavMesh). NavMesh is seen—in combination with A* search—as one of the best solutions to find the shortest path [21], and works by dividing the environment's map into polygons (termed "baking" the NavMesh). Specifically, the algorithm starts with a polygon around the current position of an agent, then adds additional polygons (in regions of the map that can be reached from existing polygons) to the set until the goal location is also contained

in one. At that point, a path to the goal is available. A* is then used to choose the best path towards the goal through the overall set of polygons. Figure 2 illustrates the NavMesh of the human agents in the obstacles environment (left), and a path created with the A* algorithm using the NavMesh (right).

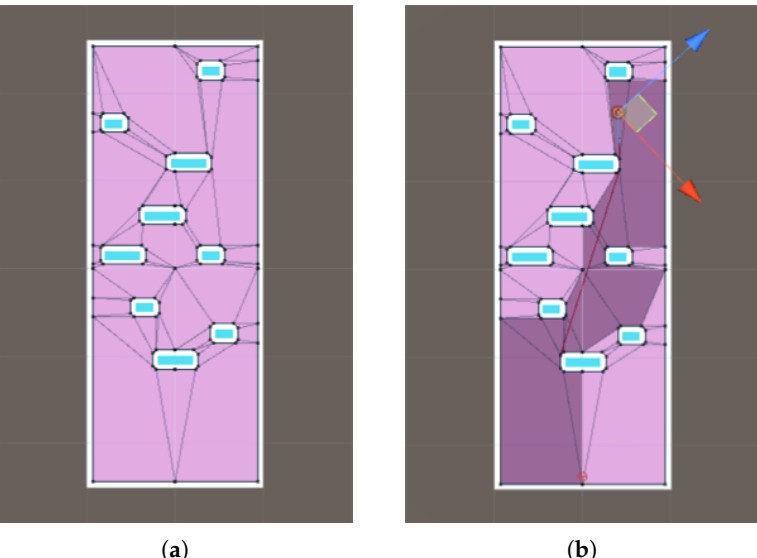

(**a**)                                                                 (**b**)

**Figure 2.** Examples of using NavMesh and A*: (**a**) example NavMesh of the obstacles environment and (**b**) example navigation plan for a simulated human agent given its current location. Dark polygons represent the path to the goal at the bottom centre. The red line shows the current planned path and the blue line the current heading of the agent. As the agent continues, the path is updated.

### 2.4. Robot Access to Information about the Environment

Our main manipulation concerns the amount of information about the environment the robot has access too. We model the robot's sensory range as a right-angled triangular region emanating from the robot position, and vary the height of this triangle (i.e., the distance it covers from the robot's location). Only information about objects within this range is taken into account by the robot in its planning activities (see Figure 3). The sensor range can take values between 5 and 30 world units in increments of 5. The full length of the environment is slightly under 60 world units; a sensor range of 30 thus covers slightly more than half the entire area ahead.

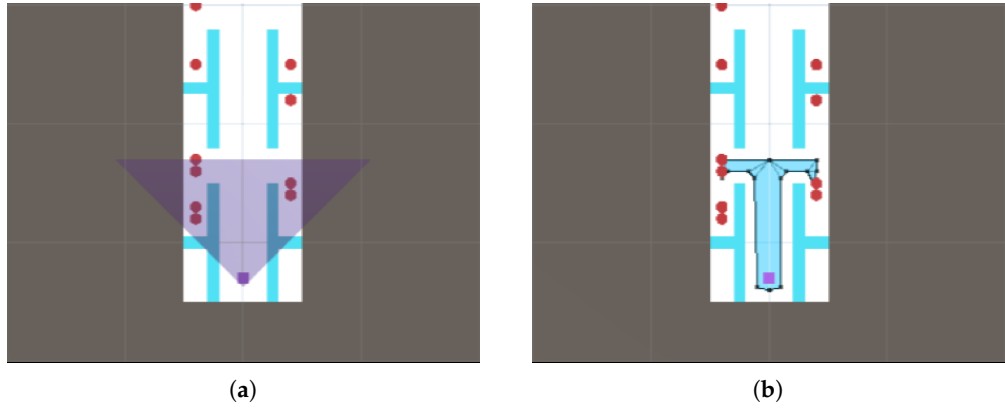

(**a**)                                                                 (**b**)

**Figure 3.** Robot sensor range example: (**a**) Triangle indicating what is perceived with a sensor range of 10 units in the office environment and (**b**) the corresponding NavMesh used by the robot.

## 2.5. Experiment

For each environment, we manipulate (1) the sensor range of the robot and (2) the "population level", that is, the number of human agents in the environment. The sensor range is increased from localist to covering just over half the distance in increments of 5 as described previously (Section 2.4).

The number of human agents, meanwhile, is determined based on the number of valid spawn points in each environment. As each environment is different, this number also differs between environments. In total, we generate five different population levels for each environment. The median population level for each level is simply set to one third of the number of spawn points. Higher and lower population levels are then determined by increasing and decreasing this value twice. For the two open environments, the step value of these increments/decrements is set to 10. For the office environment, given its smaller size, we instead used a step value of 5. The resulting population levels are summarised in Table 1, whereas sensor range and population levels explored in each environment. Further, example navigations in all three environments, with various sensor ranges indicated, can be seen in the Supplementary Materials Video S1.

**Table 1.** Different size levels of the human populations in each environment. In each case, the median value corresponds to 1/3 of valid spawn points for human agents in the corresponding environment.

| Environment | Population Levels | | | | |
|---|---|---|---|---|---|
| Office | 15 | 20 | 25 | 30 | 35 |
| Street | 80 | 90 | 100 | 110 | 120 |
| Obstacles | 75 | 85 | 95 | 105 | 115 |

We ran each possible combination 50 times, resulting in 6 sensor levels × 5 human population levels × 3 environments = 4500 runs. For each run, we stored the time it took the robot to reach the goal, as well as the number of collisions with human agents and number of collisions with static objects.

In addition to these runs, we ran a baseline in which the robot had perfect knowledge of the entire environment to have a metric by which to assess the impact of limited sensory information. This resulted in another 50 repetitions × 5 levels of humans × 3 environments = 750 runs.

## 2.6. Data Analysis

All statistics have been done in R. We test for significant effects on (1) the time to complete the navigation task, (2) the number of collisions and (3) differences with the baseline. Potential outliers are identified using the Bonferroni outlier test, visually inspected and removed if warranted (see Section 3.1). Assumptions of homogeneity (using Levene's test) of variances and normality of residuals (using Shapiro–Wilk's test) are checked following outlier removal. If both assumptions are met, a parametric two-way ANOVA test is used for tests (1) and (2), and t-tests are used for test (3). If at least one of the assumptions is not met, a nonparametric Friedman test (in case of test (1) and (2)) or Wilcoxon rank-sum test (for test (3)) is performed. The third test further involves multiple comparisons; these are adjusted for using the Bonferroni correction.

## 3. Results

### 3.1. Outlier Analysis

Initial visual inspection of the results revealed occasional potential outliers; thus, we verify whether or not those data points should actually be considered outliers using the Bonferroni outlier test. We find one significant outlier in the office environment and ten for the obstacle environment. In all cases, the cause was identified as unforeseen minor issues in the simulation: in the office environment, the robot could inadvertently enter an office and get stuck. In the obstacle environment, the outlier analysis identified an issue with the NavMesh: a small passage past one obstacle exists but if the robot actually chooses this passage, the NavMesh occasionally subsequently fails to bake the

small passage, causing the robot to get stuck until it is randomly pushed away by a wandering human agent (as also indicated by the fact that collision numbers for these outliers are quite high).

### 3.2. Robot Performance

The main results for time to completion, after outlier removal, are shown in Figure 4. For the office environment, we find no significant effect of the sensor range on the completion time (Friedman test: $p = 0.57$). However, we do find significant effects for both the open environment (Friedman test: $p < 0.001$) and the obstacle environment (Friedman test: $p < 0.01$). In both cases, the time to completion improves with increased sensor range. As expected, we also find a significant effect of the number of human agents in the environment, with more humans leading to higher completion times. Most interestingly, we find that none of the observed improvements are linear, we observe diminishing returns as sensor ranges increase if there is a return at all.

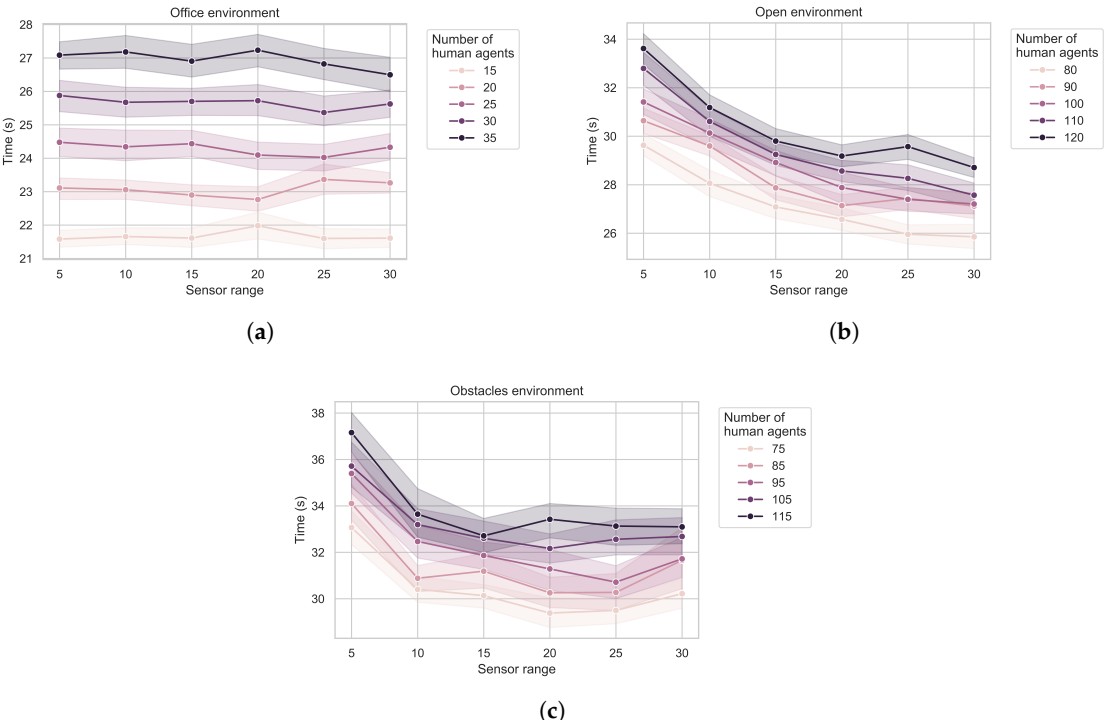

**Figure 4.** Mean time and 95% confidence intervals (shaded area) taken by the robot to complete navigation in the (**a**) office, (**b**) open, and (**c**) obstacles environment as a function of sensor range and number of humans.

When it comes to the number of collisions, we only find a significant effect in the obstacle environment (Friedman test: $p < 0.01$, whereas $p > 0.3$ in the office environment and $p > 0.27$ in the open environment), interestingly suggesting that the number of collisions increases slightly with the sensor range (see Figure 5, noting in particular the low number of collisions to start with as well as the confidence intervals).

**Table 2.** *p*-values from Wilcoxon rank-sum tests comparing ideal navigation times from the baseline to the performance summarised in Figure 4. *nH* is shorthand for *number of simulated human agents*. Significant values after Bonferroni correction are highlighted in grey.

| Env. | *nH* | Sensor Range | | | | | |
| | | 5 | 10 | 15 | 20 | 25 | 30 |
|---|---|---|---|---|---|---|---|
| Office | 15 | 0.80 | 0.48 | 0.62 | 0.15 | 0.87 | 0.74 |
| | 20 | 0.11 | 0.17 | 0.43 | 0.93 | 0.06 | 0.03 |
| | 25 | 0.26 | 0.91 | 0.59 | 0.52 | 0.59 | 0.84 |
| | 30 | 0.12 | 0.26 | 0.16 | 0.27 | 0.91 | 0.29 |
| | 35 | 0.15 | 0.13 | 0.38 | 0.06 | 0.48 | 0.63 |
| Open | 80 | $2.9 \times 10^{-16}$ | $4.2 \times 10^{-11}$ | $8.9 \times 10^{-7}$ | $5.5 \times 10^{-5}$ | $7.18 \times 10^{-3}$ | 0.09 |
| | 90 | $3.7 \times 10^{-15}$ | $2.1 \times 10^{-13}$ | $4.0 \times 10^{-5}$ | 0.02 | $1.79 \times 10^{-3}$ | 0.026 |
| | 100 | $4.7 \times 10^{-15}$ | $5.6 \times 10^{-10}$ | $3.2 \times 10^{-5}$ | 0.20 | 0.67 | 0.934 |
| | 110 | $4.0 \times 10^{-14}$ | $2.6 \times 10^{-9}$ | $6.4 \times 10^{-5}$ | $7.7 \times 10^{-3}$ | 0.13 | 0.951 |
| | 120 | $4.1 \times 10^{-16}$ | $8.8 \times 10^{-11}$ | $1.1 \times 10^{-5}$ | $4.2 \times 10^{-3}$ | $2.2 \times 10^{-4}$ | 0.071 |
| Obstacle | 75 | $1.8 \times 10^{-12}$ | $7.7 \times 10^{-6}$ | $4.5 \times 10^{-5}$ | 0.016 | $4.6 \times 10^{-3}$ | $1.2 \times 10^{-4}$ |
| | 85 | $6.7 \times 10^{-14}$ | $1.8 \times 10^{-5}$ | $2.9 \times 10^{-5}$ | $4.1 \times 10^{-3}$ | 0.020 | $7.7 \times 10^{-5}$ |
| | 95 | $7.2 \times 10^{-11}$ | $1.2 \times 10^{-4}$ | $1.0 \times 10^{-3}$ | 0.096 | 0.29 | 0.01 |
| | 105 | $2.7 \times 10^{-11}$ | $3.3 \times 10^{-5}$ | $1.2 \times 10^{-3}$ | $6.5 \times 10^{-3}$ | $1.6 \times 10^{-3}$ | $2.0 \times 10^{-3}$ |
| | 115 | $2.2 \times 10^{-11}$ | 0.01 | 0.09 | $5.5 \times 10^{-3}$ | 0.06 | 0.03 |

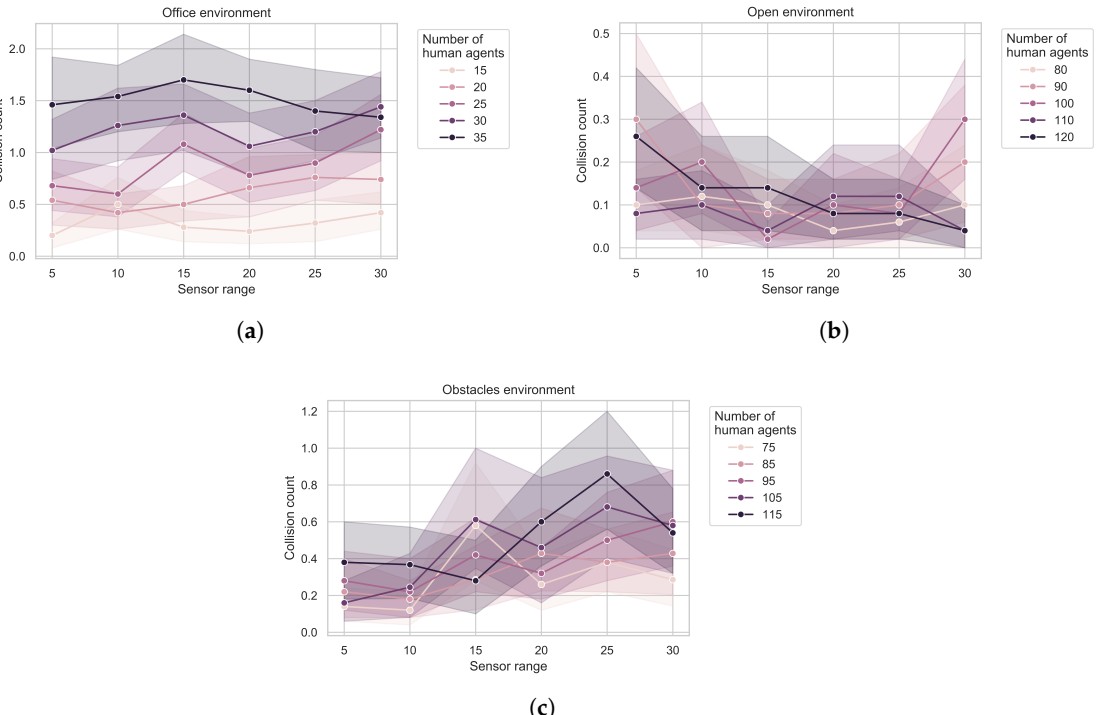

(a)

(b)

(c)

**Figure 5.** Mean and 95% confidence intervals (shaded area) of the collision counts during navigation in the (**a**) office, (**b**) open and (**c**) obstacles environment as a function of sensor range and number of humans.

### 3.3. Baseline Comparison

To put the results described above into context, we lastly compare performance with a simulated agent that has complete knowledge of the environment. The performance itself can be seen in Figure 6. We compare the performance with the corresponding performances obtained in our main

result using the Wilcoxon rank-sum test (as normality assumptions are violated) and applying the Bonferroni correction for multiple comparisons (since each baseline value is compared to six previously obtained values, this sets the significance level at $p < 0.05/6 = 8.3 \times 10^{-3}$). The full results are presented in Table 2. Of particular interest are two findings: first, we find no significant differences in the office environment, and second, although there are significant differences in the other two environments, the magnitude of the difference decreases rapidly with increasing sensor range. This further demonstrates the diminishing returns observed in Figure 4.

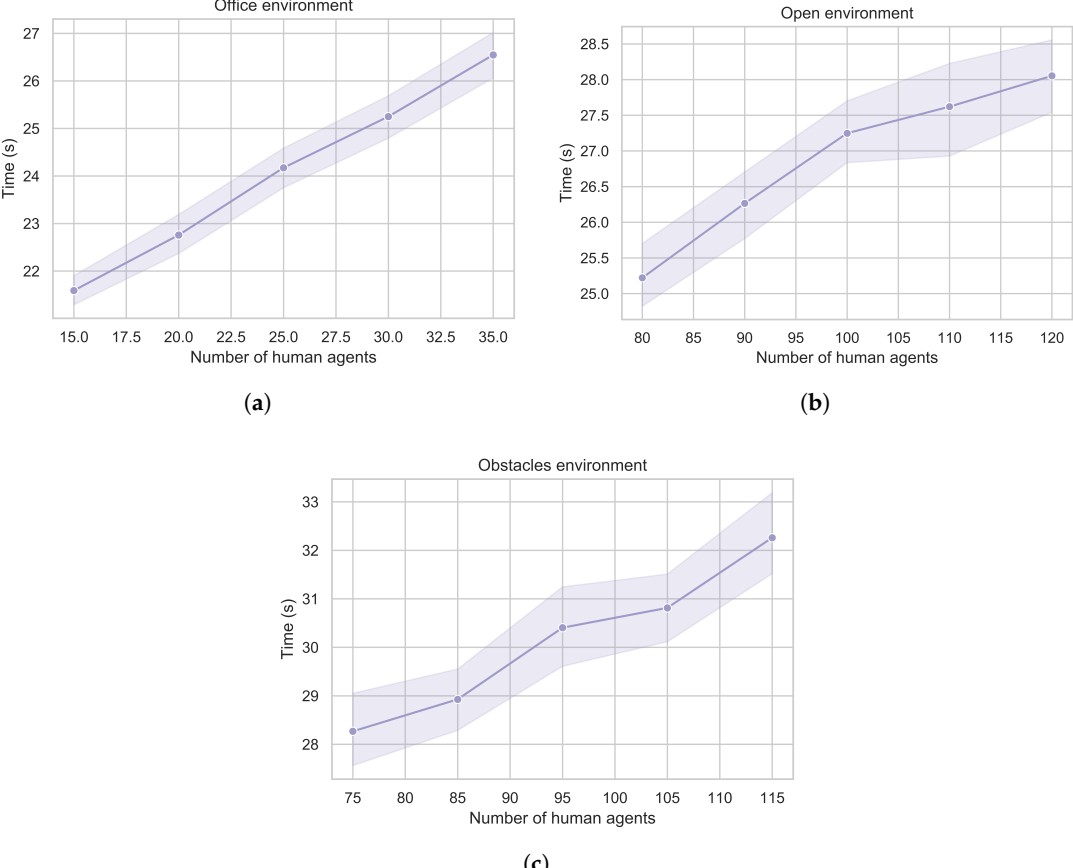

**Figure 6.** Mean and 95% confidence intervals (shaded area) of the time to completion in the (**a**) office, (**b**) open and (**c**) obstacles environment as a function of number of human agents in the environment for robots with full knowledge of the environment. In most cases, these are not significantly different from performances with a sensor range of 30 reported in Figure 4. Details of how this optimal performance compares with limited sensors are reported in Table 2.

## 4. Discussion

Overall, our results show an effect of sensor range on navigation when the navigation takes place in an open environment. In a narrow environment, like our office corridor, the navigational choices are so restricted that there is no added benefit to being able to perceive further along the route. The benefit is also stronger in open environments with obstacles than in open environments without. These results are in line with expectations and essentially demonstrate that, at least in environments where the robot has a choice of paths, being able to plan ahead for static objects has a benefit but planning ahead for dynamic objects that are almost certainly going to change position is less relevant.

However, we also find that the benefits of additional range diminish rapidly: in a fully open environment, even a sensor range that covers approximately half the total distance provides no additional benefit to being able to perceive the entire environment. Even when there is a significant

benefit, it remains small in terms of relative improvement (see Table 2) except for the more restrictive sensors we modelled. Our results therefore support the idea that limited sensory range already achieves relatively efficient performance, at least in this task.

Most interestingly, the performance penalty does not decrease linearly with increased sensor range; rather the relationship exhibits diminishing returns. For crowd navigation itself, the result demonstrates, as mentioned, that it can be achieved efficiently based on relatively local information, at least for a basic navigation strategy that ignores aspect such as social conventions in crowds [11,16]. More generally, our results thus demonstrate that it is worth first evaluating, in simulation, whether or not the added benefits obtained from a sophisticated sensing arrangement are likely to be in relation to the additional costs it entails. As such, our approach is in line other studies that investigate algorithmic strategies in simulation first, including existing work on navigation [11,17,18]; we merely use simulations to explore a different aspect of the overall robotic system.

There are some limitations that deserve discussing. The first is that the entire demonstration is carried out in simulation only, which might be overly simplifying matters. In particular, our study is missing a direct validation in hardware: primarily, there is no guarantee that the relationship between sensor range and task performance would follow the same curve as we find here. That said, other studies have shown that crowd navigation is possible using local information only [8], which at least strongly suggests that there would be limited additional benefits to making larger ranges accessible to sensors.

We also make relatively strong assumptions on the ability to interpret sensory data, as we assume that everything within the range of the robot's sensors is known to the robot. In real environments, obtaining this kind of knowledge from sensor data is not trivial [7,8], in particular for longer ranges, although a way to address this would be by mimicking work on cooperative systems in the vehicle domain [9].

A further limitation is that our simulation is not a particularly sophisticated model of crowd navigation: as already mentioned, the simulated robot does not observe any social conventions (save for avoiding collisions) and human agents all move towards a goal position using the same A*/NavMesh navigation strategy. Although this is an effective method for planning shortest paths, more sophisticated navigation algorithms could take human trajectories into account, including predictions thereof [12,22]. It is possible that a wider sensing horizon would be beneficial in that case. However, it would still remain the case that trajectories closer to the robot are more relevant than those further away.

Note that our simulated robot with the A*/NavMesh navigation strategy does actually succeed in the navigation task with a minimal number of collisions. This at least raises the question whether crowd navigation would necessarily benefit substantially from more sophisticated algorithms. In our case, two factors primarily contributed to the good performance: one was that human agents would also try to avoid collisions and not otherwise attempt to sabotage the robot (whereas interaction with real humans might require active steps on the part of the robot to avoid intentional sabotage by humans [23]). The second was that a simulated robot is able to react quickly, both in terms of sensing and acting. For present day robots, quick reactive changes in direction may still be a challenge and anticipation of future obstacles therefore beneficial. Overall, it is thus possible that further development in terms of robot mechanics might alleviate a need for sophisticated planning in this type of scenario in the future.

Finally, a curious result we found concerned a slight increase in collision with increased sensor range in one of our environments. We do not have a good interpretation of this. Considering that the effect is relatively small, the most likely explanation is that it is just a spurious result. This is further supported by the relatively large confidence intervals for the means obtained.

## 5. Conclusions

To conclude, we have investigated how reducing the range for which a robot has access to information about its environment affects its ability to successfully navigate through crowds in different types of environments. We demonstrate diminishing returns of increased sensory sophistication if such a return exists at all: the rate at which performance can be improved reduces as more resources are spent on this improvement. This suggests that future work can benefit from exploring added benefits of various sensory strategies in simulation before embarking on a real-world implementation. For crowd navigation, work in simulation is abundant but focuses largely on navigation algorithms; here, we show that investigating the sensing aspect is in itself useful. Simulations are increasingly used in the machine learning community [24,25] and in robot control [26], where they were long discouraged. Here, we have provided a proof of concept of the utility of even relatively simple simulations in investigating questions pertinent to social robotics.

**Supplementary Materials:** The following are available online at http://www.mdpi.com/2078-2489/11/2/112/s1, Video S1: Example navigations in all three environments with various sensor ranges indicated.

**Author Contributions:** Conceptualisation, M.H. and S.T.; methodology, M.H. and S.T.; formal analysis, M.H.; investigation, M.H.; writing—original draft preparation, M.H.; writing—review and editing, S.T.; visualisation, M.H.; supervision, S.T. All authors have read and agreed to the published version of the manuscript.

**Funding:** This research received no external funding.

**Conflicts of Interest:** The authors declare no conflicts of interest.

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
