# Peer review of "How Much Information Does a Robot Need? Exploring the Benefits of Increased Sensory Range in a Simulated Crowd Navigation Task"

_information, doi:10.3390/info11020112_

Round 1

Reviewer 1 Report

This paper investigates a very important issue in navigation and the structure of the paper is relatively complete. The paper should make a comprehensive summary of the current research of solving the problem of balancing  the sensing sophistication and performance. Otherwise, the method proposed later in the paper lacks a certain foundation.

Author Response

Dear reviewer,

The paper should make a comprehensive summary of the current research of solving the problem of balancing the sensing sophistication and performance.

thank you for this comment. We do fully agree; however, as far as we can tell, there is not much in terms of direct investigations relating sensor sophistication to task performance in the literature, so there is also not much to summarise. We have instead added a not to that effect.

In part, it is also worth highlighting that our use of the word "sophistication" might have been slightly confusing - we have clarified this now in the manuscript (and also by updating the title). We are specifically concerned with sensor range (local information being accessible with on-robot sensors, while global information would require a sensorised environment). There are of course examples in the literature, where cheap sensors are compared to expensive ones, but this is not the manipulation we wanted to address here (because it doesn't change the range of the available information; only its quality). This question of "how much of the environment do I need to sense?" appears not overly explored, in particular in hardware. We are however happy to be pointed to studies on this that we might have missed.

Another way to address the point is to note that different studies sometimes solve the same task with different hardware setups. While it is not possible to compare these directly for performance (because of different setups, etc), it does at least give some indication for the range of choices that could in principle be used in a given task. For crowd navigation in particular, both local and global sensing solutions can be found. This was already in the paper, but the point did not come across clearly; we have now rewritten that part of the introduction.

Reviewer 2 Report

The authors present a study based on the law of diminishing returns applied to sensory sophistication.

The work is developed in different simulation environments and the conclusions of this work are supported by the experimental results.

I recommend a general deep review rewriting some parts in terms of clarity, i.e., Abstract, Conclusions... Many sentences are not clear (lines 5-7, 52-57, 93-98,....)

Author Response

Dear reviewer,

thank you for pointing these out - we have rephrased sentences throughout. The new version highlights all changes (identified using latexdiff), except those to the abstract and title. We believe the text is now easier to follow (in particular, we have also simplified some of the argumentation to focus more directly on sensor range as opposed to somewhat conflating range with cost).

Round 2

Reviewer 1 Report

The overall structure of the revision is more complete and optimized. The analysis section could be further enhanced,especially, supplementing the schematic representation of the relationships between analysis factors.

Author Response

Dear reviewer,

thank you for pointing us to the analysis section. We agree that this was somewhat confusing, but we think it was also at least in part due to a sub-optimal description of the experimental setup before. Also, table 2 does summarise the various combinations of runs, but only appears relatively late in the manuscript.

We have therefore:

  • rewritten both the experimental description and the description of the data analysis, in particular simplifying the latter since the relevant parts of the results section provide the necessary context.
  • added a reference to table 2 to help the reader.